# Bayesian Symbolic Regression with Entropic Reinforcement Learning

## Abstract

Symbolic regression is the problem of finding an algebraic expression describing a stochastic dependence of a target variable on a set of inputs. Unlike forms of regression that fit parameters assuming a fixed model structure, symbolic regression is a search problem over the space of expressions, represented, for example, as abstract syntax trees using a library of operators. Symbolic regression is typically used in settings with limited, noisy data in the natural sciences. However, searching for a single best-fitting expression fails to capture the epistemic uncertainty about the expression, which motivates a Bayesian perspective that enables uncertainty quantification and specification of natural priors to constrain the search space. In this work, we propose ERRLESS (Entropy-Regularised Reinforcement Learning for Expression Structure Sampling), a scalable approach for sampling the posterior distribution over expressions given data using maximum-entropy reinforcement learning. ERRLESS learns a neural policy that constructs expressions sequentially by building up their abstract syntax trees. At convergence, the policy samples expressions from the posterior. At test time, expressions can be sampled by rollouts of this policy. We demonstrate that ERRLESS achieves near state-of-the-art exact symbolic recovery on the AI Feynman benchmark (Udrescu & Tegmark, 2020). Beyond exact recovery, we demonstrate that the mean of the posterior predictive approximated by ERRLESS achieves a coefficient of determination ($R^2$) of 0.98, highlighting the benefits of the Bayesian perspective in symbolic regression.

## 1 Introduction

Symbolic regression (SR) is the problem of searching over a space of algebraic expressions, using a certain library of primitive operators, to find a function that most closely maps the inputs observed in a dataset to their corresponding outputs. SR is a common problem in the natural sciences, where datasets are small and noisy, domain priors constrain plausible formulas, and interpretability is important (Bongard & Lipson, 2007; Schmidt & Lipson, 2009; Udrescu & Tegmark, 2020). Most existing algorithms for SR (Petersen et al., 2019; Biggio et al., 2021; Mundhenk et al., 2021; Tenachi et al., 2023; Kamienny et al., 2023) have the goal of finding a *single* best expression. With limited data, such a point estimate can be unreliable and hides the uncertainty about the expression arising from the noise and scarcity of data.

The need to model uncertainty in SR has been recognized in the literature and addressed by a Bayesian perspective on the problem (Jin et al., 2019; Guimera & Sales-Pardo, 2025). In this view, one posits a (structured) prior over expression structures and parameters and a model of observation noise. A dataset of (input, output) pairs then induces a posterior distribution over expressions, and the aim of Bayesian SR is to sample from this posterior. Previous Bayesian SR methods use reversible-jump Markov chain Monte Carlo (MCMC) (Green, 1995) or sequential Monte Carlo (SMC) (Bomarito & Leser, 2025; Guimera & Sales-Pardo, 2025), but these methods rely on handcrafted proposal distributions and can be costly to scale. In this paper, we instead seek an approach that amortizes the sampling process using a neural network.

We formulate the construction of an expression as a sequential decision-making problem, turning the Bayesian SR task into the reinforcement learning (RL) problem of training a policy to sample expressions from the posterior. To achieve unbiased sampling at convergence, we train this policy using a maximum-entropy RL objective with the unnormalised posterior log-density as the reward

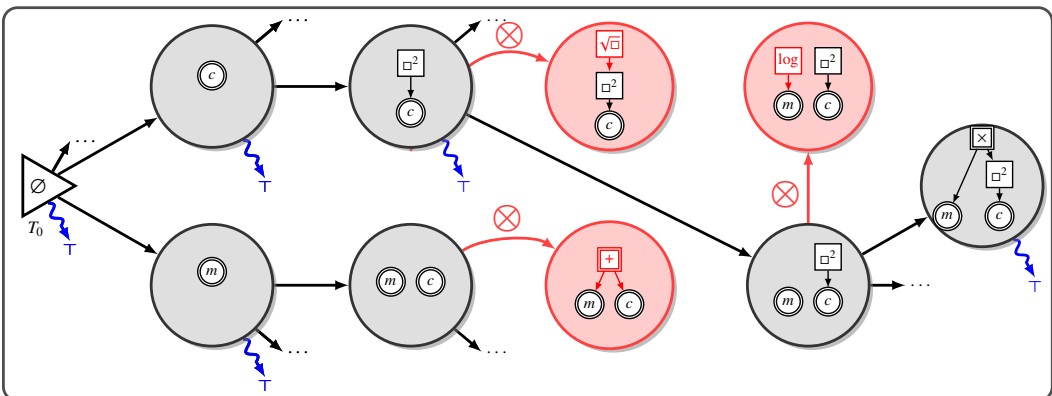

Figure 1: **Generation process for the expression** $E = mc^2$. Valid states are shown in gray, while invalid states are highlighted in red. Black arrows indicate valid transitions, and pink arrows indicate invalid transitions. Leaf nodes are represented by double-outlined circles, unary operators by single-outlined squares, and binary operators by double-outlined squares. Wiggly arrows denote states that can terminate and transition to the terminal state.

(§3.2). The resulting system, which we call ERRLESS (Entropy-Regularised Reinforcement Learning for Expression Structure Sampling), is capable of learning a policy that samples expression structures and their real-valued parameters; once trained, the policy can be sampled to produce approximate posterior samples efficiently.

Successfully applying entropy-regularized RL to Bayesian SR requires careful design choices. ERRLESS uses a generation process that constructs expression syntax trees in a bottom-up (postorder) manner to allow effective imposition of constraints, imposes structural priors to avoid ill-formed, redundant, or unlikely subexpressions, and can incorporate additional constraints to restrict the search space to forbid dimensionally incompatible compositions of physical units (§2.1). The parametrization and training of the policy similarly require appropriate design of neural architectures and off-policy training schemes (§3.3). Finally, unlike previous Monte Carlo-based methods, ERRLESS amortizes sampling of both expression structures and parameter values into a single neural policy and avoids explicit per-candidate constant fitting.

On the Feynman Symbolic Regression Database (Udrescu & Tegmark, 2020), ERRLESS achieves a competitive exact symbolic recovery rate and produces posterior samples that improve predictions when data are few and noisy. Beyond exact recovery, the posterior predictive mean reaches a median $R^2$ of 0.98 on the noisiest setting, showing the value of modeling uncertainty over expressions with a Bayesian view.

We summarize our contributions as follows:

- We design a novel generation process (environment) for symbolic regression that enables bottom-up expression construction, incorporates dimensional analysis, and imposes structural constraints.
- We formulate Bayesian symbolic regression, including inference of scalar parameters in expressions, as an end-to-end policy-learning problem within this environment.
- We demonstrate that ERRLESS achieves competitive exact symbolic recovery compared to state-of-the-art approaches on the Feynman Symbolic Regression Database.
- We show that our approach captures the posterior distribution effectively, facilitating downstream applications in settings with scarce and noisy data.

## 2 BAYESIAN SYMBOLIC REGRESSION

**Notation.** Scalar (resp. vector) random variables, or variables whose type has not been specified, are denoted x (resp. **x**), and their realization is denoted $x$ (resp. $\boldsymbol{x}$). The probability distribution of a discrete random variable x is denoted $P_x$ (in uppercase) and that of a continuous random variable, or a variable whose type has not been specified, is denoted $p_x$ (in lowercase). Let $\mathbb{N}$ (resp. $\mathbb{R}$) be the set of natural (resp. real) numbers, and $\mathbb{N}_{>0} := \mathbb{N} \setminus \{0\}$ (resp. $\mathbb{R}_{>0} := \{x \in \mathbb{R} \mid x > 0\}$). For $n \in \mathbb{N}$,

we denote the set of integers $\{1, \ldots, n\}$ (or the empty set when $n = 0$) as $[n]$. For a set $S$, we denote by $|S|$ its cardinality and by $S^* := \bigcup_{n \in \mathbb{N}} S^n$ the set of all finite sequences (strings) over $S$.

## 2.1 The Space of Expressions

Let $D, K, M \in \mathbb{N}$. For ease of understanding, we will first define the space of *expression trees*, ignoring physical unit constraints, then modify it to include these constraints.

**Expression trees.** We fix a vocabulary $\Sigma := \mathcal{V} \sqcup C \sqcup O$, where $\mathcal{V} := \{v_i\}_{i \in [D]}$ is a set of *variable symbols*, $C := \{c_k\}_{k \in [K]}$ is a set of *constant symbols*, and $O := \{g_j\}_{j \in [M]}$ is a set of *operator symbols* ($O$ is referred to as *operator library*). Each operator $g \in O$ is equipped with an *arity* $r_g \in \mathbb{N}_{>0}$ and a *semantic function* $\phi_g : \mathcal{X}_g \to \mathbb{R}$, where $\mathcal{X}_g \subseteq \mathbb{R}^{r_g}$. For example, the operator '+' is binary and $\phi_+ : \mathbb{R}^2 \to \mathbb{R}, (x, y) \mapsto x + y$; the operator 'log' is unary and $\phi_{\log} : \mathbb{R}_{>0} \to \mathbb{R}, x \mapsto \log x$.

An *expression tree* T is a finite, rooted, ordered tree where each leaf is labeled with a symbol from $\mathcal{V} \sqcup C$, and each internal node with $r$ children is labeled with a $g \in O$ such that $r_g = r$. Let $I_C(\mathrm{T}) \subseteq [K]$ (resp. $I_{\mathcal{V}}(\mathrm{T}) \subseteq [D]$) be the set of indices of constants (resp. variables) appearing in T. The constants $\{c_k\}_{k \in I_C(\mathrm{T})}$ are thought of as symbolic placeholders: a *constant assignment* is a vector $\boldsymbol{\theta} \in \mathbb{R}^K$ that specifies numerical values for these symbols at indices $I_C(\mathrm{T})$.

Let $\mathcal{T}_\Sigma$ be the space of expression trees over $\Sigma$.

**Evaluation.** Let $\mathrm{T} \in \mathcal{T}_\Sigma$. Given a constant assignment $\boldsymbol{\theta} := (\theta_k)_{k \in K} \in \mathbb{R}^K$, one can define a partial *evaluation function* $f_{\mathrm{T}, \boldsymbol{\theta}} : \mathbb{R}^D \to \mathbb{R}$ for T on every input $\boldsymbol{x} := (x_i)_{i \in D} \in \mathbb{R}^D$ recursively as follows:

- each leaf of T labeled $v_i \in \mathcal{V}$ (resp. $c_k \in C$), where $i \in I_{\mathcal{V}}(\mathrm{T})$ (resp. $k \in I_C(\mathrm{T})$), evaluates to $x_i$ (resp. $\theta_k$);
- each internal node of T labeled with operator $g$ having children that evaluate to $a_1, \ldots, a_{r_g}$ evaluates to $\phi_g(a_1, \ldots, a_{r_g})$ if $(a_1, \ldots, a_{r_g}) \in \mathcal{X}_g$, otherwise the evaluation is undefined.

**Sequence representation.** An expression tree T can be canonically represented as a sequence $\mathcal{W}(\mathrm{T}) \in \Sigma^*$ by listing the labels of its nodes in postorder traversal, a representation also known as reverse Polish notation (Łukasiewicz, 1929). For example, the expression $\sin(v_1 + c_1 \times v_2)$ is represented by the sequence $(v_1, c_1, v_2, \times, +, \sin)$. (It is well-known that such an expression can be evaluated, when real numbers are substituted for the variables and constants, by traversing the sequence from left to right and maintaining a stack.)

**Unit constraints.** One can additionally constrain the space of trees by modifying the above definitions to include dimension information. We assume that each variable $v_i$ has an associated physical unit (*e.g.*, meters, kilograms), represented as a vector $\boldsymbol{u}_i \in \mathbb{R}^\ell$ (called *unit vector*), where $\ell := 7$ is the number of base units.[1] Each coordinate of $\boldsymbol{u}_i$ represents the exponent of its corresponding unit, *e.g.*, velocity has units $\mathrm{m\,s}^{-1}$, represented as $(1, 0, -1, 0, 0, 0, 0)$. Dimensionless variables are represented by the zero vector $(0, \ldots, 0) \in \mathbb{R}^\ell$. All constants $c_k$ are treated as dimensionless.

Every operator $g$ is assumed to have constraints on the unit vectors of its arguments, as well as a mapping from the unit vectors of the arguments to the unit vector of its output. Formally, every $g$ has a *unit assignment function*, a partial function $\mathcal{U}_g : (\mathbb{R}^\ell)^{r_g} \to \mathbb{R}^\ell$ that returns the unit vector of the operator's output if the operator can be applied to inputs with the given units (and is undefined otherwise). Let $\mathcal{T}^\circ \subseteq \mathcal{T}_\Sigma$ be the set of those expression trees which are dimensionally valid: when evaluating bottom-up, $\mathcal{U}_g$ is never undefined when evaluated on the unit vectors of the children of any internal node. See §A.1 for the assignment functions of operators in the library we use.

---

**Example.** Consider the following operators applied to $\boldsymbol{u}^{\mathrm{vel}} := (1, 0, -1, 0, 0, 0, 0)$, $\boldsymbol{u}^{\mathrm{time}} := (0, 0, 1, 0, 0, 0, 0)$, and $\boldsymbol{u}^{\mathrm{angle}} := (0, 0, 0, 0, 0, 0, 0)$:

- +: $\mathcal{U}_+(\boldsymbol{u}^{\mathrm{vel}}, \boldsymbol{u}^{\mathrm{time}})$ is undefined (incompatible units: time and velocity cannot be added), but $\mathcal{U}_+(\boldsymbol{u}^{\mathrm{vel}}, \boldsymbol{u}^{\mathrm{vel}}) = \boldsymbol{u}^{\mathrm{vel}}$ (adding two velocities gives a velocity);
- ×: $\mathcal{U}_\times(\boldsymbol{u}^{\mathrm{vel}}, \boldsymbol{u}^{\mathrm{time}}) = (1, 0, 0, 0, 0, 0, 0)$ (multiplying velocity and time gives a distance);
- sin: $\mathcal{U}_{\sin}(\boldsymbol{u}^{\mathrm{angle}}) = \boldsymbol{u}^{\mathrm{angle}}$ (sine of a scalar is a scalar), but $\mathcal{U}_{\sin}(\boldsymbol{u}^{\mathrm{vel}})$ is undefined.

---

[1] In physics, the International System of Units is based on seven base units: meter (m), kilogram (kg), second (s), ampere (A), kelvin (K), mole (mol), and candela (cd).

## 2.2 BAYESIAN SYMBOLIC REGRESSION

We fix a prior probability distribution $P(\mathrm{T})$ over the set of valid expression trees $\mathcal{T}^\circ$ and a conditional prior distribution $p(\boldsymbol{\theta} \mid \mathrm{T})$ over the space $\mathbb{R}^K$ of constant assignments.

Let $\{(\mathbf{x}_i, \mathbf{y}_i)\}_{i=1}^N$ be a collection of random variables such that, for every $i \in [N]$, $(\mathbf{x}_i, \mathbf{y}_i)$ takes values in $\mathbb{R}^D \times \mathbb{R}$, and each realization $\boldsymbol{x}_i \in \mathcal{X}$ of $\mathbf{x}_i$ defines the following conditional distribution:

$$\mathbf{y}_i \mid (\mathbf{x}_i = \boldsymbol{x}_i, \mathrm{T}, \theta) \sim \mathcal{N}\big(f_{\mathrm{T},\boldsymbol{\theta}}(\boldsymbol{x}_i), \sigma^2\big), \tag{1}$$

where $\sigma^2 \in \mathbb{R}_{>0}$ is a fixed noise variance and the $\mathbf{y}_i$'s are conditionally independent given the $\mathbf{x}_i$'s. That is, $\mathbf{y}_i$ is the evaluation of the expression given by $\mathrm{T}, \boldsymbol{\theta}$ with inputs $\boldsymbol{x}_i$, with added Gaussian noise.

We observe a dataset $\mathcal{D} := \{(\boldsymbol{x}_i, y_i)\}_{i=1}^N$. The posterior of $(\mathrm{T}, \boldsymbol{\theta})$ given $\mathcal{D}$ satisfies the following relation:

$$p(\mathrm{T}, \boldsymbol{\theta} \mid \mathcal{D}) \propto p(\mathrm{T}, \boldsymbol{\theta}) \cdot p(\mathcal{D} \mid \mathrm{T}, \boldsymbol{\theta});$$

$$\propto p(\mathrm{T}, \boldsymbol{\theta}) \prod_{i=1}^N p(y_i \mid \boldsymbol{x}_i, \mathrm{T}, \boldsymbol{\theta}) \underbrace{p(\boldsymbol{x}_i \mid \mathrm{T}, \boldsymbol{\theta})}_{= \, p(\boldsymbol{x}_i)}, \quad (\boldsymbol{x}_i\text{'s are independent of T and } \boldsymbol{\theta});$$

$$\propto p(\mathrm{T}, \boldsymbol{\theta}) \prod_{i=1}^N \exp\left(-\frac{\big(y_i - f_{\mathrm{T},\boldsymbol{\theta}}(\boldsymbol{x}_i)\big)^2}{2\sigma^2}\right), \quad (\text{by (1)}). \tag{2}$$

The objective of Bayesian symbolic regression is to sample from the posterior distribution. Having formulated the mathematical setting, in the next sections, we introduce the framework used to approximate this distribution in practice.

# 3 METHODOLOGY

We introduce ERRLESS (Entropy-Regularised Reinforcement Learning for Expression Structure Sampling), a scalable approach for Bayesian symbolic regression, illustrated in Fig. 1. §3.1 introduces a sequential decision-making process for sampling expression trees that enforces constraints during generation, §3.2 describes learning objectives for decision-making policies, and §3.3 describes the policy architecture used to sample expressions.

For concreteness, we specify the set of operators we will use in our experiments: the unary operators $\big\{\sin, \cos, \log, \exp, \square^2, \sqrt{\square}, -\square\big\}$ and the binary operators $\{+, -, /, \times\}$. However, the algorithm described below is not restricted to this particular choice.

## 3.1 BOTTOM-UP GENERATION

Most approaches in the literature (Petersen et al., 2019; Li et al., 2023) employ *top-down generation*, where internal nodes (*i.e.*, operators) are sampled before leaf nodes (input variables and constants). With top-down generation, the intermediate expression at each step contains 'holes' to be filled. As a result, the expression can only be evaluated, and the posterior density computed, at the end of the generation. Moreover, with top-down generation, enforcing physical unit constraints is inefficient: since the operands may not yet be specified at intermediate steps, assigning units requires traversing the full depth of the tree once the units have been determined.

In contrast, *bottom-up generation* offers two main advantages. First, some intermediate states are valid and complete expressions. Second, because leaf nodes (and therefore operands) are specified from the outset, newly added operators can be constrained to be compatible with the known physical units of the operands.

**Sequential generation of expression trees.** Recall that an expression tree T is uniquely represented by its postorder representation $\mathcal{W}(\mathrm{T}) \in \Sigma^*$. Therefore, generating an expression tree is equivalent to generating its sequence representation by appending one symbol at a time from left to right. Because all operator nodes appear after their arguments in postorder traversal, the dimensional validity and arity constraints correspond to restrictions on continuations of partial sequences. The problem of modeling a probability distribution over expression trees is thus reduced to one of autoregressive sequence modeling.

We define the token alphabet $\mathcal{A} := \Sigma \sqcup \{\top\}$, where $\top$ is a special symbol marking sequence termination. A distribution over the set of trees $\mathcal{T}^\circ$ is equivalent to a distribution over $\mathcal{A}^*$:

$$\pi(w_1 w_2 \ldots w_n \top) = \pi(w_1)\pi(w_2 \mid w_1) \ldots \pi(w_n \mid w_1 \ldots w_{n-1})\pi(\top \mid w_1 \ldots w_n), \quad w_i \in \mathcal{A}, \quad (3)$$

where the support of each next-symbol distribution respects the unit and arity constraints, as well as the restriction that generation of $\top$ is permitted only after a sequence that represents a complete expression tree. (We must also make the assumption that every partial sequence can be continued to a sequence ending in $\top$, which holds with the library of operators we consider.)

**Length and redundancy constraints.** We remark that constraints on the number of nodes in the expression tree T can also be expressed as restrictions on the support of the next-token distributions of $\mathcal{W}(\mathrm{T})$ in (3). In addition, structural constraints in the target distribution that disallow ill-formed, redundant, or unlikely compositions can also be expressed as restrictions on next-token distributions. The constraints we impose prohibit: (i) composing functions with their inverses (*e.g.*, $\sqrt{\square}^2$), (ii) nesting trigonometric functions (*e.g.*, $\sin(\cos(\square))$), (iii) nesting exponentials (*e.g.*, $e^{e^\square}$), and (iv) applying unary operators directly to constants. Fig. 2 shows how our construction rules and constraints make the search space much smaller and thus more efficient to explore.

**Samplers of expressions and parameters as policies.** A distribution $\pi$ over sequences of the form (3) respecting the imposed constraints, together with a conditional distribution $\pi(\boldsymbol{\theta} \mid \mathrm{T})$ over constant assignments given a tree, define a joint distribution over $(\mathrm{T}, \boldsymbol{\theta})$. In the next section, we will describe how this autoregressively factorized distribution can be trained to sample the Bayesian posterior defined in §2.2 using reinforcement learning methods.

### 3.2 Maximum-entropy RL training of expression samplers

Above, we have identified a distribution $\pi$ over pairs $(\mathrm{T}, \boldsymbol{\theta})$ with a distribution over sequence representations and a conditional distribution over constant assignments:

$$\pi(\mathrm{T}, \boldsymbol{\theta}) = \pi(\mathrm{T})\pi(\boldsymbol{\theta} \mid \mathrm{T}) = \pi(\mathcal{W}(\mathrm{T})\top)\pi(\boldsymbol{\theta} \mid \mathrm{T}),$$

where $\pi(\mathcal{W}(\mathrm{T})\top)$ has an autoregressive factorization (3). Suppose that $\pi$ is a parametric model $\pi_\varphi$, which can be evaluated to obtain next-token logits for sequential generation of $\mathcal{W}(\mathrm{T})\top$ and the parameters of the distribution over $\boldsymbol{\theta}$ given T (for example, the mean and covariance of a Gaussian from which $\boldsymbol{\theta}$ is sampled). Our goal is to fit $\varphi$ so that $\pi_\varphi(\mathrm{T}, \boldsymbol{\theta})$ equals the posterior $p(\mathrm{T}, \boldsymbol{\theta} \mid \mathcal{D})$ defined in (2), which is given as an unnormalized probability density function.

Define

$$R(\mathrm{T}, \boldsymbol{\theta}) = \log p(\mathrm{T}, \boldsymbol{\theta}) + \log \prod_{i=1}^N \exp\left(-\frac{(y_i - f_{\mathrm{T}, \boldsymbol{\theta}}(\boldsymbol{x}_i))^2}{2\sigma^2}\right),$$

so that $p(\mathrm{T}, \boldsymbol{\theta} \mid \mathcal{D}) \propto \exp(R(\mathrm{T}, \boldsymbol{\theta}))$. When $R$ is thought of as a *reward* provided to a sampler that generates T and $\boldsymbol{\theta}$ following the conditional distributions of $\pi_\varphi$, training $\pi_\varphi$ to maximize its expected reward $\mathbb{E}_{(\mathrm{T}, \boldsymbol{\theta}) \sim \pi_\varphi}[R(\mathrm{T}, \boldsymbol{\theta})]$ is equivalent to minimizing cross-entropy between $\pi_\varphi$ and the posterior, which is achieved by sampling the posterior mode. However, one can instead consider the *entropy-regularized* reinforcement learning problem

$$\max_\varphi \left[\mathbb{E}_{(\mathrm{T}, \boldsymbol{\theta}) \sim \pi_\varphi}[R(\mathrm{T}, \boldsymbol{\theta})] + \mathcal{H}[\pi_\varphi]\right], \quad (4)$$

where $\mathcal{H}[\pi_\varphi]$ is the entropy of the modeled distribution. A key property of entropy-regularized, or maximum-entropy RL (Haarnoja et al., 2018; Eysenbach & Levine, 2022) is that the solution to (4) minimizes KL divergence between $\pi_\varphi$ and the distribution with density proportional to $\exp(R(\mathrm{T}, \boldsymbol{\theta}))$, *i.e.*, the posterior. This property has been exploited in various instances of learning to sample by sequential decision-making (Deleu et al., 2024), and efficient off-policy training algorithms for solving (4) have been proposed.

**Training objective.** One such off-policy objective is *trajectory balance* (TB) objective (Malkin et al., 2022), denoted by $\mathcal{L}_{\mathrm{TB}}$, which is a special case of a path consistency learning objective (Nachum et al., 2017) in deterministic environments with sparse terminal rewards. TB requires additionally learning a scalar parameter $\log Z_\varphi$ (corresponding to the initial state's value function in reinforcement learning), which, at convergence, gives the normalizing constant (the likelihood of $\mathcal{D}$). The objective associated with $(\mathrm{T}, \boldsymbol{\theta})$ is:

$$\mathcal{L}_{\mathrm{TB}}(\mathrm{T}, \boldsymbol{\theta}; \varphi) := \left[\log Z_\varphi + \log \pi_\varphi(\mathrm{T}, \boldsymbol{\theta}) - R(\mathrm{T}, \boldsymbol{\theta})\right]^2. \quad (5)$$

Because (5) can be minimized to 0 for *all* samples simultaneously, training algorithms can optimize this objective w.r.t. $\varphi$ over $(T, \theta)$ sampled from some behavior policy that does not necessarily coincide with the current state of $\pi_\varphi$ itself. We describe our off-policy training choices in §3.3. (Because the reward can equal 0, we apply smoothing to prevent $\log 0$ in the loss; see §A.2.)

### 3.3 ERRLESS DESIGN CHOICES

**Parametrization of the policy.** We represent each partial expression as a sequence of tokens and parameterize the policy with a transformer (Vaswani et al., 2017). The transformer encodes the sequence into an embedding that is supplied to two prediction heads: (i) a head that produces the logits over the next action at each step of the construction process, for a sequence that has not terminated in $\top$, and (ii) a head that outputs the mean and variance of a Gaussian distribution from which the values of the constants are sampled, for a sequence that has terminated (see §B for more details).

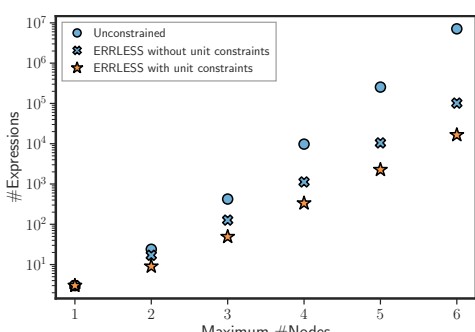

Figure 2: Growth of search space size in maximum number of nodes (in log-scale).

**Prior and tempering.** We choose a tempered unigram prior $P(T)$ over expression trees at temperature (see §A.2 for more details) together with length constraints and the redundancy constraints described in §3.1. For the prior $p(\theta \mid T)$ over constant assignments, we chose the uniform prior over the interval $[0, 10]$, with a penalty for the number of constants used. Additionally, to encourage our policy to sample small expressions, we use a soft-length prior (Landajuela et al., 2021) centered around 8 with variance 5 (see §B.3 for more details).

**Training policy.** To encourage exploration, we use off-policy training and use $\epsilon$-greedy exploration with annealed $\epsilon$ along with a prioritized replay buffer. See §B for all relevant hyperparameters and training details.

## 4 RELATED WORKS

**Symbolic regression with deep learning.** Deep symbolic regression (Petersen et al., 2019, DSR;) trains an autoregressive recurrent neural network (RNN) policy via risk-seeking policy gradient algorithms. Tenachi et al. (2023) uses the same learning algorithm but enforces constraints on physical units at each step of the generation. ERRLESS differs from both approaches by taking a Bayesian perspective using maximum entropy RL to train the policy and using a bottom-up generative process. Mundhenk et al. (2021) combine DSR with genetic programming, resulting in improved exploration and recovery of benchmark formulas. Beyond RL, NeSymReS (Biggio et al., 2021) is a transformer pre-trained on an equation corpora, yielding zero-shot generalization across diverse symbolic regression tasks. Kamienny et al. (2023) integrates a pre-trained model within Monte Carlo Tree Search (MCTS). Contrary to these approaches ERRLESS does not rely on existing datasets and adopts the Bayesian perspective on symbolic regression.

**Bayesian symbolic regression and probabilistic modeling.** The method Bayesian Symbolic Regression (BSR; Jin et al., 2019) uses reversible-jump MCMC (Green, 1995) to sample expressions from the posterior distribution. Bomarito & Leser (2025) replaces MCMC with sequential Monte Carlo (SMC). Finally, Guimera & Sales-Pardo (2025) provides a statistical physics perspective on Bayesian symbolic regression. Unlike existing BSR methods that rely on handcrafted proposal distributions and computationally intensive MCMC sampling, ERRLESS learns an amortized posterior sampler with maximum entropy reinforcement learning.

**Sampling discrete structured posteriors with maximum-entropy reinforcement learning.** ERRLESS builds upon prior work on maximum entropy RL for sampling from discrete structured distributions (Buesing et al., 2020; Bengio et al., 2021). The trajectory balance objective (Malkin et al., 2022), equivalent to path consistency learning (Nachum et al., 2017), has been used for posterior inference over decision trees (Mahfoud et al., 2025), causal models (Deleu et al., 2022; 2023), phylogenetic trees (Zhou et al., 2024). GFN-SR (Li et al., 2023) uses trajectory balance to learn policies for sampling expressions on small synthetic benchmarks. ERRLESS deviates from GFN-

Table 1: **Posterior predictive metrics on the synthetic dataset.** We report the accuracy of the mean of the posterior predictive ($R^2_{PP}$) as well as the negative log-likelihood (NLL) on the test set for all noise levels. The results are the median across the datasets and the seeds.

| Noise level → | $\gamma = 0.001$ | | $\gamma = 0.01$ | | $\gamma = 0.1$ | |
|---|---|---|---|---|---|---|
| Algorithm ↓ Metric → | $R^2_{PP}$ | NLL | $R^2_{PP}$ | NLL | $R^2_{PP}$ | NLL |
| **ERRLESS** | **0.99** | $-222.23$ | **0.99** | **$-222.10$** | **0.99** | **$-225.03$** |
| **Parallel-tempering MH** | -4.164275 | **$-223.60$** | 0.97 | $-221.77$ | $-1.11e+11$ | $-109.29$ |
| **Gaussian Process** | $-0.30$ | $-15.80$ | $-0.31$ | $-16.26$ | $-0.40$ | $-40.85$ |

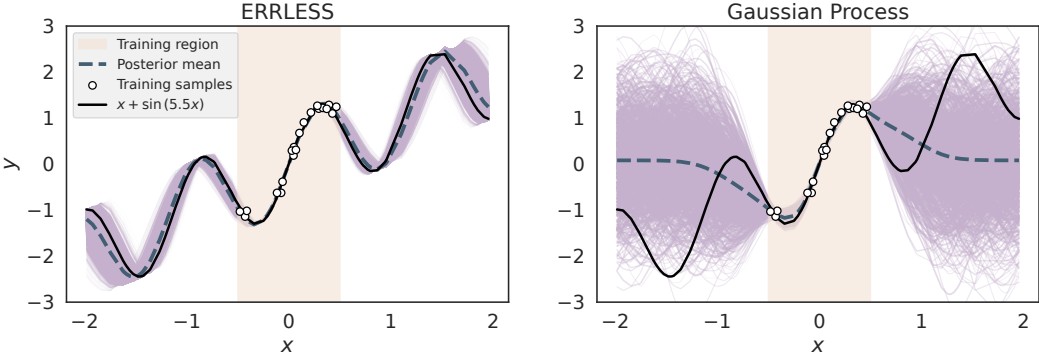

Figure 3: **Samples from the posterior**. ERRLESS and a Gaussian process are trained on noised values (white circles) of the ground-truth function (solid black line) at points in the training domain (beige). The posterior mean (dashed blue line) – the mean of the individual posterior samples (purple) – fits the true function well on the beige interval in both cases, but ERRLESS extrapolates better outside the training region.

SR using a better generative process for constructing expressions, incorporating explicit priors, and evaluation on large-scale benchmarks complemented by improved training.

## 5 EXPERIMENTAL SETUP

In this section, we describe our experimental setup. We compare our approach against the baselines from La Cava et al. (2021) and PhySO (Tenachi et al., 2023), a state-of-the-art symbolic regression method. A detailed description of these baselines is provided in Appendix §B.5. In §5.1, we briefly summarize the benchmark datasets, and in §5.2, we outline the evaluation metrics.

### 5.1 DATASETS

**Synthetic dataset.** We construct a small dataset of five expressions to closely study the quality of posterior modeling. Each expression has 20 points for training and 100 points for testing. We set the maximum number of nodes to $L = 6$ and the maximum number of constants to $K = 1$. The expressions contain at most two variables. We give an overview of the expressions and their corresponding datasets in §B.1.

**Feynman Symbolic Regression Database.** We use the Feynman Symbolic Regression Database (Udrescu & Tegmark, 2020), a standard benchmark for symbolic regression comprising 120 physics-inspired expressions originating from the *Feynman Lectures on Physics* (Feynman et al., 2015). Following Tenachi et al. (2023), we remove 4 expressions involving arccos or arcsin, leaving 116 expressions in total[2]. Each expression is paired with 1M sampled points; we subsample 10,000 for training (to compute the reward) and 25,000 for testing, as per the SRBench protocol (La Cava et al., 2021). We set $L = 35$ and $K = 3$.

The benchmark datasets are originally noise-free. Following the protocol of La Cava et al. (2021), we add Gaussian noise to the ground-truth targets in the train set only. The noise is sampled from

---

[2]The expressions removed are: `feynman_I_26_2`, `feynman_I_30_5`, `feynman_II_11_17` and `feynman_test_10`

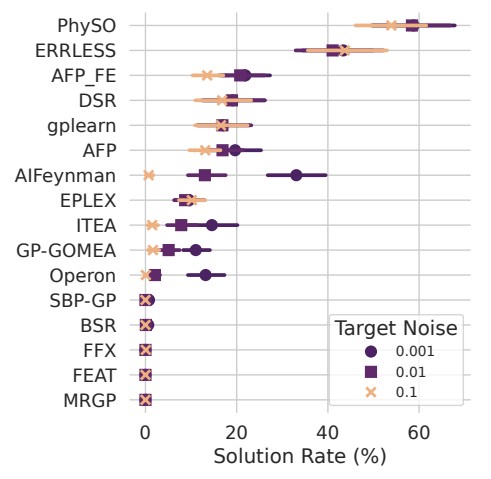

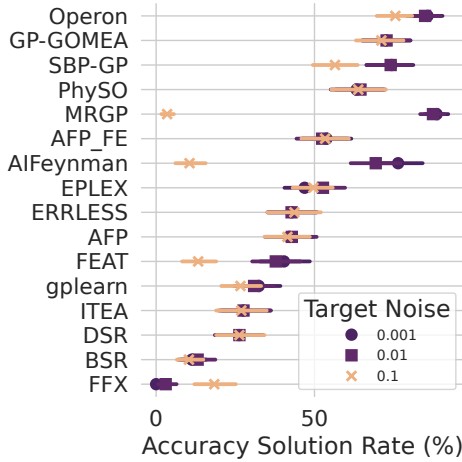

(a) Exact symbolic recovery rate

(b) Rate of accuracy ($R^2 > 0.999$)

Figure 4: **Results on the Feynman Symbolic Regression Database:** (a) Exact symbolic recovery rate and (b) rate of accuracy.

$\mathcal{N}\left(0, \gamma \sqrt{\frac{1}{N} \sum_{i=1}^{N} y_i^2}\right)$, where $\gamma$ controls the noise level. For each expression, we run our algorithm with five random seeds and three noise settings: $\gamma \in \{0.001, 0.01, 0.1\}$ on the same training split. Importantly, the noise is sampled once per dataset and kept fixed across all runs to ensure comparability.

## 5.2 METRICS

Let $Y = \{y_i\}_{i=1}^{N}$ (resp. $\hat{Y} = \{\hat{y}_i\}_{i=1}^{N}$) denote the target (resp. prediction) and their mean $\bar{Y}$ (resp. $\bar{\hat{Y}}$).

**Prediction accuracy.** We use the coefficient of determination $R^2$ as our accuracy metric between the target $Y$ and prediction $\hat{Y}$:

$$R^2 := 1 - \frac{\sum_{i=1}^{N} (y_i - \hat{y}_i)^2}{\sum_{i=1}^{N} (y_i - \bar{Y})^2}.$$

The coefficient $R^2 \le 1$ quantifies the fraction of variance in the target $Y$ explained by the model: $R^2 = 1$ indicates a perfect fit, $R^2 = 0$ indicates performance equivalent to predicting the mean of $Y$, and $R^2 < 0$ is worse than that baseline.

**Exact symbolic recovery (La Cava et al., 2021).** Given a predicted expression tree T with parameters $\theta$ and a ground-truth tree T* with parameters $\theta^*$: $(T, \theta)$ is considered equivalent to $(T^*, \theta^*)$ if $f_{T,\theta}$ does not reduce to a constant and either $f_{T,\theta}/f_{T^*,\theta^*}$ reduces to a non-zero constant or $f_{T,\theta} - f_{T^*,\theta^*}$ reduces to a constant[3].

**Rate of accuracy (La Cava et al., 2021)** Given a predicted expression tree T with parameters $\theta$: $(T, \theta)$ is said to be accurate if the test set $R^2$ of $f_{T,\theta}$ is strictly bigger than 0.999.

## 6 RESULTS

### 6.1 POSTERIOR MODELING OVER SMALL EXPRESSIONS

We evaluate posterior modeling and compare ERRLESS against a Gaussian process on the synthetic datasets introduced in §5.1. We draw 1000 samples from the posterior for both methods. Table 1 shows that our approach accurately models the posterior over expression trees, as indicated by the accuracy of the posterior predictive mean ($R^2_{PP}$) and the negative log-likelihood (NLL). In Figure 3, we can see that ERRLESS produces posterior samples that fit the data even though the training region is small enough not to reveal the true shape of the function. The posterior predictive mean mirrors the ground-truth function, and the posterior samples fit the train set accurately while only diverging slightly from the function on the test set.

---

[3]Intuitively, this criterion accounts for equivalent expressions that differ only by a multiplicative or additive constant, ensuring that structurally equivalent formulas are recognized even if they are scaled or shifted.

## 6.2 DISCOVERING PHYSICS FORMULAS

**Posterior modeling.** We draw 1000 samples from our approximate posterior to compute the posterior predictive (see §C). We use a cold posterior at temperature $\frac{1}{2}$ (*i.e.*, raising the modeled distribution to a power, cf. Zhang et al. (2018); Wenzel et al. (2020)). For noise level $\gamma = 0.1$, the mean posterior predictive of ERRLESS attains a median $R^2$ of 0.98 on the Feynman Symbolic Regression Database, and 0.97 for $\gamma \in \{0.01, 0.001\}$. These results show that our method is effective for downstream scientific tasks where it is important to have an ensemble of models explaining the data and combining their predictions to obtain uncertainty estimates as well.

**Exact symbolic recovery and fit quality.** For $\gamma = 0.001$, ERRLESS attains an exact symbolic recovery rate of 44.31%, compared to 58.78% for the state-of-the-art PhySO. On the rate of accuracy metric – defined as the proportion of expressions with $R^2 > 0.999$ – ERRLESS reaches 42.76%, while PhySO achieves 63.30%. As shown in Fig. 4, our method is notably more robust to noise, exhibiting only minor variations in both symbolic recovery and accuracy across different noise levels. Furthermore, ERRLESS significantly outperforms BSR, a Bayesian symbolic regression approach, in exact symbolic recovery. The performance gap between ERRLESS and PhySO is partly explained by training budgets. Both methods are limited to 1M samples, but while PhySO optimizes directly for the reward, ERRLESS must approximate the posterior, a task that typically requires a larger training budget.

## 6.3 QUALITATIVE ANALYSIS

We examine expressions discovered by our sampler, and highlight the case of the ground-truth expression $\rho_0/\sqrt{1 - \frac{v^2}{c^2}}$. The highest-scoring candidate found by ERRLESS was $\frac{\rho_0}{\cos(v/c)}$. Although these two expressions appear unrelated at first glance, their denominators have the same second-order Taylor approximation in $\frac{v}{c}$ at 0. The length of the postorder representation of the ground-truth expression is greater than that of its approximate counterpart, which is unfavored by the prior. An extreme example of the same is the ground-truth expression $0.159 h\omega / \left(\exp\left(0.159 \frac{h\omega}{Tk_B}\right) - 1\right)$, which has the first-order Taylor approximation $Tk_B$. The prior strongly favors $Tk_B$, and the two expressions have similar likelihood: the linear approximation achieves a test set $R^2$ of 0.99.

These results can be explained by the fact that ERRLESS incorporates a prior that favors concise expressions, as opposed to methods that only optimize the quality of fit. With more data samples, we would expect the likelihood to eventually dominate the prior in the reward, and the ground-truth expression would have a higher score.

## 7 CONCLUSION

We introduced ERRLESS, a scalable approach to Bayesian symbolic regression, using maximum-entropy reinforcement learning to amortize posterior sampling over algebraic expressions describing a stochastic dependence of a target variable on its inputs. We formulate expression synthesis as a sequential decision-making process and prune the search space by enforcing dimensional constraints during bottom-up construction. On the Feynman Symbolic Regression Database (Udrescu & Tegmark, 2020), ERRLESS achieves an exact symbolic recovery rate competitive with state-of-the-art methods such as PhySO (Tenachi et al., 2023), while approximating the full posterior distribution, rather than returning a single-point estimate. The posterior predictive mean achieves strong accuracy (median $R^2 = 0.98$), demonstrating robustness in noisy regimes. Such posterior samples enable interpretable model discovery, hypothesis aggregation, and uncertainty-aware extrapolation beyond the training support, all of which are key to scientific discovery.

**Limitations and future work.** While our method accurately models the posterior over expressions, performance can degrade on long or highly complex target expressions. Future work could condition the sampler on the temperatures of the log-likelihood and the priors, to better trade off complexity and accuracy. Additionally, amortizing the sampler over datasets would allow posterior sampling at inference time without the need for retraining. Symbolic regression methods that amortize over datasets have already shown promise in optimizing for the best expression given a dataset (Biggio et al., 2021; Kamienny et al., 2023). This naturally motivates extending ERRLESS to learn the operator library itself, allowing for re-usable constructs that recur across datasets (Ellis et al., 2021).

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

# Appendix

## Table of Contents

## A  BOTTOM-UP GENERATION

### A.1  OPERATOR PHYSICAL UNIT CONSTRAINTS AND ASSIGNMENTS

We show the values of the partial *unit assignment* function for the operators that we use in our library in Table 2.

| Operator $g$ | Unit assignment function $\mathcal{U}_g$ |
|:---:|:---:|
| $+$ | $\mathcal{U}_+(u_1, u_2) = u_1$ if $u_1 = u_2$, else undefined |
| $-$ | $\mathcal{U}_-(u_1, u_2) = u_1$ if $u_1 = u_2$, else undefined |
| $\times$ | $\mathcal{U}_\times(u_1, u_2) = u_1 + u_2$ |
| $/$ | $\mathcal{U}_/(u_1, u_2) = u_1 - u_2$ |
| $\sin$ | $\mathcal{U}_{\sin}(u) = 0$ if $u = 0$, else undefined |
| $\cos$ | $\mathcal{U}_{\cos}(u) = 0$ if $u = 0$, else undefined |
| $\log$ | $\mathcal{U}_{\log}(u) = 0$ if $u = 0$, else undefined |
| $\exp$ | $\mathcal{U}_{\exp}(u) = 0$ if $u = 0$, else undefined |
| $\square^2$ | $\mathcal{U}_{\square^2}(u) = 2u$ |
| $\sqrt{\square}$ | $\mathcal{U}_{\sqrt{\square}}(u) = \frac{1}{2}u$ |
| $-\square$ | $\mathcal{U}_{-\square}(u) = u$ |

Table 2: Physical unit assignment function for the full operator library. When a variable is dimensionless, its unit vector is $0$.

### A.2  REWARD FUNCTION

**Unigram prior over expressions.** Following Constantin et al. (2024), mathematical formulas are known to obey Zipf's law. We leverage this observation by constructing a unigram prior over operators based on their empirical frequencies. Specifically, we use *Encyclopaedia Inflationaris* as the reference corpus, extract operator frequencies (Table 1 in their paper), discard operators not included in our library, and renormalize the distribution. The resulting frequencies for our operator set are shown in Table 3.

Table 3: Unigram prior over operators, variables, and constants. Frequencies are normalized after restricting to our operator library.

| Token | Frequency |
|:---|:---:|
| $*$ | 0.1770 |
| $/$ | 0.1328 |
| $-$ | 0.0476 |
| $+$ | 0.0454 |
| $\square^2$ | 0.0365 |
| $\exp$ | 0.0210 |
| $\sqrt{\square}$ | 0.0199 |
| $-\square$ | 0.0177 |
| $\log$ | 0.0133 |
| $\cos$ | 0.0072 |
| $\sin$ | 0.0048 |
| Variables | 0.2877 |
| Constants | 0.1892 |

**Choice of $\sigma$ for training.** In practice, using the same $\sigma$ used to generate data in the log-likelihood is impractical and yields a log-likelihood that is very large in magnitude, which introduces numerical instabilities during the training. Let $\sigma'$ be the new alternative standard deviation to be used in the log-likelihood, the pair $(T^*, \theta^*)$ with the highest log-likelihood $\ell_{\max}$, and a pair $(T, \theta)$ with a log-

likelihood $\ell_0$. Let $\Delta_{\max} = \ell_0 - \ell_{\max}$, then we can express it as:

$$\Delta_{\max} = \ell_0 - \ell_{\max}$$

$$= \frac{N}{2\sigma'^2}\left[\text{MSE}(T, \boldsymbol{\theta}) - \text{MSE}(T^*, \boldsymbol{\theta}^*)\right]$$

Where $\text{MSE}(T, \boldsymbol{\theta}) = \frac{1}{N}\sum_{i=1}^{N}\left(y_i - f_{T,\boldsymbol{\theta}}(\boldsymbol{x}_i)\right)^2$ is the mean-squared error. We can now solve for $\sigma'$:

$$\sigma'^2 = \frac{N}{2\Delta_{\max}}\left[\text{MSE}(T, \boldsymbol{\theta}) - \text{MSE}(T^*, \boldsymbol{\theta}^*)\right] \tag{6}$$

We choose $\Delta_{\max} := 200$ and $\text{MSE}(T, \boldsymbol{\theta}) = \left(1 - R^2_{\min}\right)\text{Var}(T^*, \boldsymbol{\theta}^*)$, where $\text{Var}(T^*, \boldsymbol{\theta}^*)$ denotes the variance of the ground-truth outputs generated by $(T^*, \boldsymbol{\theta}^*)$ on the training set, and we choose $R^2_{\min} := 0.3$.

With this definition, any candidate expression $(T, \boldsymbol{\theta})$ whose $R^2$ score falls below $R^2_{\min}$ is automatically assigned a log-likelihood smaller than $-\Delta_{\max} = -200$. To avoid excessively large penalties, we clamp all log-likelihood values to a minimum of $-200$.

When the evaluation of an expression with given input $\boldsymbol{x}$ occurring in the computation of the reward for $(T, \boldsymbol{\theta})$ fails due to the presence of an operator with inputs outside of its domain (for example, when evaluating $\sqrt{c_1 \cdot v_1}$ with $c_1 = -5$ and $v_1 = 2$), the minimum reward is also automatically assigned.

# B EXPERIMENTAL DETAILS

## B.1 DATASETS

**Synthetic.** Table 4 shows the five expressions we used for benchmark alongside the ranges used for the training and test set, respectively.

Table 4: **Synthetic dataset**. Variables are sampled uniformly from the specified intervals.

| Expression | Training range | Test range |
|---|---|---|
| $x + \sin(5.5x)$ | $x \sim \mathcal{U}(-0.5, 0.5)$ | $x \sim \mathcal{U}(-2, 2)$ |
| $4.567 \cdot e^x + y$ | $x, y \sim \mathcal{U}(-1, 1)$ | $x, y \sim \mathcal{U}(-2, 2)$ |
| $\sin(y + 2.5x)$ | $x, y \sim \mathcal{U}(0, 1)$ | $x, y \sim \mathcal{U}(0, 3)$ |
| $\sqrt{x^2 + y^2}$ | $x, y \sim \mathcal{U}(0, 1)$ | $x, y \sim \mathcal{U}(0, 3)$ |
| $\sin(x)\cos(y)$ | $x, y \sim \mathcal{U}(0, 1)$ | $x, y \sim \mathcal{U}(0, 3)$ |

**Feynman Symbolic Regression Database.** We show characteristics of the Feynman Symbolic Regression Database (Udrescu & Tegmark, 2020) (see Fig. 5). In particular: (i) the distribution over the number of variables in a given expression in the dataset, (ii) the distribution over expression lengths of a given expression in the dataset, and (iii) the ratio of expressions with physical units compared to unitless ones.

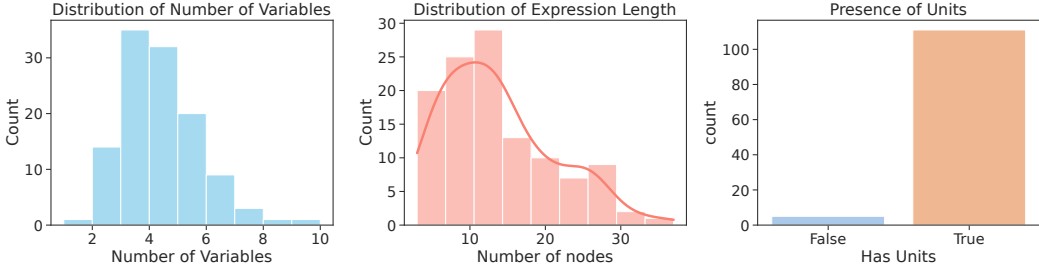

Figure 5: **Exploratory analysis of the Feynman Symbolic Regression Database**. Left: histogram of the number of variables per expression. Center: distribution of expression tree lengths. Right: proportion of expressions in which all variables are associated with physical units.

**Blackbox benchmark** contains a set of 122 datasets intended for benchmarking machine learning methods from PMLB (Romano et al., 2021). These datasets contain categorical features as well as categorical target variables. While SRBench report results on all 122 datasets, we only consider a subset of 66 datasets that are purely meant for regression and thus only contain continuous features and targets.

**Strogatz benchmark** The Strogatz benchmark were taken from the ODE-Strogatz repository (La Cava et al., 2016). Each dataset in the benchmark is a 2D coupled Ordinary Differential Equation (ODE) where the goal is to predict the rate of change of the 2D variable. The benchmark contains 7 ODE that represent natural processes.

## B.2 Replay buffer

We use a modified version of the prioritized replay buffer where each expression tree T can be stored up to a maximum of $N_{\text{repeat}}$ times in the buffer. This is to ensure that the model is trained on different values of the constants for the same expression. When a new batch of elements is added, we apply a first filtering step where we only keep the ones that have a reward higher than the minimal reward in the buffer. This ensures that the minimum reward in the buffer never decreases, and that we don't hinder the quality of the samples in the buffer. After that, we perform a second filtering step where we make sure that none of the added expression trees are repeated more than $N_{\text{repeat}}$ times.

## B.3 Soft length prior

Landajuela et al. (2021) observe that if the initial action logits are set to zero, the resulting distribution over expression lengths is biased toward expressions whose lengths are close to the maximum allowed. To mitigate this, they propose a *soft length prior*, which introduces a negative penalty on actions likely to produce overly long expressions. Let $L$ denote the maximum number of nodes in an expression tree, and let $N_{\text{nodes}}$ be the current number of nodes. For fixed parameters $0 < \eta < L$ and $\nu \in \mathbb{R}_{>0}$, the soft length prior is implemented as an additional logit vector added to the policy logits:

- **Variable and constant actions:** Their logits are penalized according to $-\frac{(N_{\text{nodes}} - \eta)^2}{2\nu^2} \mathbb{1}_{N_{\text{nodes}} < \eta}$ which discourages adding variables or constants when the expression is still short.
- **Binary operator actions:** Their logits are penalized as $-\frac{(N_{\text{nodes}} - \eta)^2}{2\nu^2} \mathbb{1}_{N_{\text{nodes}} > \eta}$ which discourages adding binary operators once the expression grows longer than $\eta$.
- **Unary operator actions:** Their logits remain unchanged, *i.e.* equal to 0.

where $\mathbb{1}$ denotes the indicator function. The above logit is for the top-down generation case, where binary operators are responsible for growing the expression length and the variable/constants nodes for shrinking it. Since we do bottom-up generation, it's the opposite. The variable and constant nodes, by virtue of being sampled first, make the expression length larger, while binary operators do the opposite. That is why we flip the logits for binary operators and variable/constants.

## B.4 Hyperparameters

**Policy.** Recall that an expression tree T is uniquely represented by its postorder sequence $\mathcal{W}(\mathrm{T}) \in \Sigma^*$. We first embed it using a learnable embedding matrix with hidden dimension 256 to get a representation. A learned positional embedding is then added to this representation, which is processed by a Transformer encoder with 2 layers and 4 attention heads (Vaswani et al., 2017). The output of the encoder is used in two ways:

- Action selection: A linear layer maps the representation to logits over the discrete action space (operators, variables, and termination).
- Constant generation: The same representation is passed through a linear layer to produce the mean and variance of a Gaussian distribution, from which constant values are sampled.

**Off-policy training.** We use $\epsilon$-greedy policy where $\epsilon$ decays exponentially from 1 to 0.01 during the training. We use a prioritized replay buffer with a capacity of 10000 trajectories. The batch size is set to 800 with 55% of the samples in the batch coming from the replay buffer. We train the policy for a total of 1250 iterations, which corresponds to 1M visited states.

**Reward function.** The inverse temperature of the prior factors is $\alpha = 0.1$ for the expression tree log-prior and $\lambda = 0.2$ for the parameters log-prior.

**Optimization.** We use the Adam optimizer without weight decay (Kingma & Ba, 2015). The policy $\log \pi_\varphi(\mathrm{T}, \boldsymbol{\theta})$ learning rate is set to 0.001 and that of the log-partition function $\log Z_\varphi$ to 0.01.

### B.5 BASELINES

**AFP / AFP_FE (Schmidt & Lipson, 2011; 2009).** Age-Fitness Pareto optimization (AFP) is a genetic programming (GP) strategy that frames search as a bi-objective problem over prediction error and individual age. Each candidate solution is assigned a fitness value and an age, defined as the number of generations since creation. At each generation, selection operates on the Pareto front with respect to these two objectives, favoring individuals that are either accurate and relatively young or novel relative to the current population. By rewarding both accuracy and youth, AFP maintains diversity and reduces premature convergence, while still steering the search toward low-error expressions.

**AIFeynman (Udrescu & Tegmark, 2020).** AIFeynman is a physics-inspired, multi-stage method designed to discover symbolic expressions by systematically breaking down a complex problem into simpler ones. It does not learn a single generative model but rather follows a deterministic, divide-and-conquer strategy. First, a neural network is trained to high accuracy on the dataset $\mathcal{D}$. This network is then treated as an "oracle" and is probed to discover properties of the underlying function, such as symmetries, separability, or polynomial structure. Based on these discovered properties, the original problem is recursively simplified. The final, simplified sub-problems are then solved using a combination of brute-force search and polynomial fitting.

**BSR (Jin et al., 2019).** Bayesian symbolic regression (BSR) directly addresses the problem of posterior inference over the space of expressions. Similar to our approach, BSR also aims to sample from the posterior distribution $p(\mathrm{T}, \boldsymbol{\theta} \mid \mathcal{D})$, where T is the expression tree structure and $\theta$ represents its constant parameters. Due to the varying dimensionality of its parameter space (due to constantly changing tree structure), BSR employs Reversible-Jump MCMC (RJMCMC), a technique that allows the MCMC sampler to propose "moves" between models of different dimensions (e.g., adding or removing a node in the tree T) while maintaining the detailed balance condition. This approach relies on handcrafted proposal distributions for these moves, so it can be computationally intensive and often fails to explore the posterior landscape sufficiently.

**DSR (Petersen et al., 2019).** Deep symbolic regression (DSR) was a seminal work that first framed the symbolic regression task as a reinforcement learning (RL) problem. DSR employs a Recurrent Neural Network (RNN) as a policy, which autoregressively generates an expression tree T token by token in a top-down, pre-order traversal. A complete expression constitutes a trajectory, and the quality of this expression (e.g., its $R^2$ score on the dataset $\mathcal{D}$) serves as the reward $R(\mathrm{T})$. The policy is trained using a risk-seeking policy gradient algorithm, which biases the search towards high-reward expressions and helps escape local optima. The ultimate goal of DSR is to find a single best-fitting expression that maximizes the expected reward.

**PhySO (Tenachi et al., 2023).** PhySO uses the same learning framework as DSR for training their policy, but they add physical units constraints where they force the expression to be dimensionally valid at each generation step.

**EPLEX (La Cava et al., 2019a).** EPLEX is an advanced parent selection method for Genetic Programming designed to excel in continuous-valued regression tasks. Traditional selection methods rely on an aggregate fitness score (like average error), which loses information about performance on individual data points. EPLEX addresses this by filtering the population sequentially on a random ordering of individual training cases.

**FEAT (La Cava et al., 2019b).** FEAT is a hybrid method for symbolic regression that combines evolutionary computation with linear models. Instead of evolving a single monolithic expression, FEAT evolves a set of simpler expression trees that serve as features. These features are then used as inputs to a linear model. The key innovation is a feedback mechanism where the coefficients learned by the linear model are used to guide the evolutionary search, prioritizing the mutation and replacement of less impactful features.

**FFX (McConaghy, 2011).** FFX is a non-evolutionary, deterministic algorithm for symbolic regression that casts the problem as a feature selection task within a generalized linear model. The method operates in two main stages: first, it deterministically generates a massive library of candidate basis functions by applying a predefined set of nonlinear operators and interactions to the

input variables. Second, it employs path-wise regularized learning (specifically, an elastic net) to efficiently search this vast feature space.

**GP-GOMEA (Virgolin et al., 2020).** GP-GOMEA is a model-based evolutionary algorithm that aims to improve search efficiency by explicitly learning and exploiting the structure of promising solutions. Unlike traditional GP, which relies on blind genetic operators like crossover and mutation, GP-GOMEA learns a "linkage model" in each generation. This model, typically a Linkage Tree built using mutual information between nodes in the population's expression trees, identifies groups of genes (sub-programs) that work well together.

**gplearn (Stephens, 2015).** GPLearn is a Python library that implements tree-based genetic programming for symbolic regression within the `scikit-learn` API (Buitinck et al., 2013). Candidate models are mathematical expression trees evolved using standard GP operators such as sub-tree crossover and mutation, with fitness measured by prediction error.

**ITEA (de Franca & Aldeia, 2020).** ITEA is an evolutionary algorithm that operates on a constrained representation called Interaction Transformation (IT). Unlike the free-form trees in traditional GP, an IT expression is restricted to a linear combination of nonlinear terms. ITEA evolves a population of these structured expressions using a mutation-only strategy.

**MRGP (Arnaldo et al., 2014).** MRGP is a hybrid technique that integrates tree-based GP with the LASSO regularization method. Unlike conventional GP, MRGP does not directly compare the final program's output with the target variable. Instead, it constructs a set of sub-expressions from the program and fits a linear combination of these sub-expressions to the target output. The target variable is then compared against the output of the resulting regression model.

**Operon (Kommenda et al., 2019).** Operon is a modern, highly efficient C++ framework for GP in symbolic regression. It focuses on improving performance and scalability through advanced architectural choices, including representing expression trees in a cache-friendly, continuous memory layout. It also implements a fine-grained, low-overhead concurrency model for parallel execution.

**SBP-GP (Virgolin et al., 2019).** SBP-GP is a GP method that guides variation using semantic backpropagation (SB). Instead of random tree changes, SB computes the output each sub-tree should produce to help the overall expression match the target, by propagating the target values down through the tree using function inverses. New sub-trees are then generated or selected to better match these desired outputs.

## C POSTERIOR PREDICTIVE

For any test input $\boldsymbol{x}^*$, the posterior predictive distribution of the corresponding output $\mathrm{y}^*$ is given by:

$$p(\mathrm{y}^* \mid \boldsymbol{x}^*, \mathcal{D}) = \sum_{\mathrm{T}} \int p(\mathrm{y}^* \mid \boldsymbol{x}^*, \mathrm{T}, \boldsymbol{\theta}) \, p(\mathrm{T}, \boldsymbol{\theta} \mid \mathcal{D}) \, \mathrm{d}\boldsymbol{\theta} \tag{7}$$

Intuitively, this distribution weighs the contribution of all expression trees and their parameters.

**Mean of the posterior predictive.** In general, the mean is:

$$\mathbb{E}\left[\mathrm{y}^* \mid \boldsymbol{x}^*, \mathcal{D}\right] = \sum_{T} \int p(\mathrm{T}, \boldsymbol{\theta} \mid \mathcal{D}) \mathbb{E}\left[\mathrm{y}^* \mid \boldsymbol{x}^*, \mathrm{T}, \boldsymbol{\theta}\right] \mathrm{d}\boldsymbol{\theta}$$

Since the likelihood model is a Gaussian (1), we have $\mathbb{E}\left[\mathrm{y}^* \mid \boldsymbol{x}^*, \mathrm{T}, \boldsymbol{\theta}\right] = f_{\mathrm{T}, \boldsymbol{\theta}}(\boldsymbol{x}^*)$ which yields:

$$\mathbb{E}\left[\mathrm{y}^* \mid \boldsymbol{x}^*, \mathcal{D}\right] = \mathbb{E}_{\mathrm{T}, \boldsymbol{\theta}}\left[f_{\mathrm{T}, \boldsymbol{\theta}}(\boldsymbol{x}^*)\right] \tag{8}$$

In practice, we compute the above quantity by drawing expression trees with their parameters from the learned approximation to the posterior. A tempered posterior can be obtained by weighting these samples by their probability under the trained sampler raised to a power. For example, for a posterior at temperature $\frac{1}{2}$, we simply reweight the samples by their likelihood under the trained model.

## D COMPUTATIONAL RUNTIME

Figure 6 reports the runtime (in seconds) for all methods evaluated on the blackbox dataset. PhySO is omitted because its authors do not provide runtime measurements. Baseline runtimes are taken

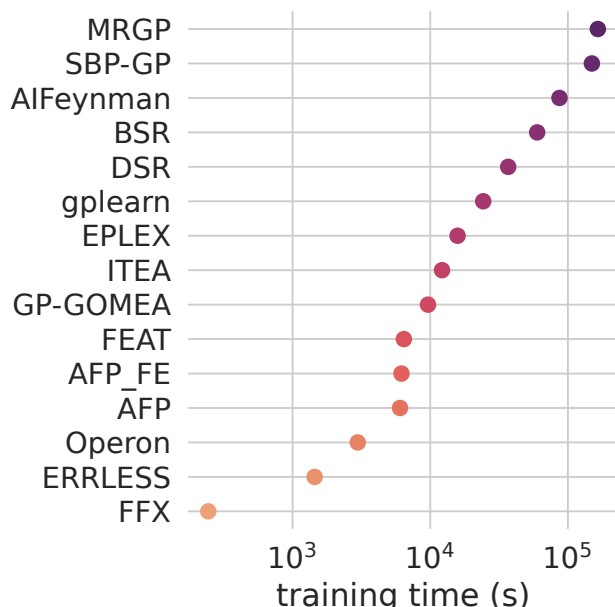

Figure 6: **Computational runtime.** Average of the time (in seconds) taken by all methods over random seeds and expressions in the Blackbox benchmark.

from SRBench. ERRLESS was run on a machine equipped with an L40S GPU, 4 CPUs, and 48 GB of RAM.

From the figure, we observe that our method is among the fastest. While hardware differences and code optimization must be considered for a fully fair comparison, we emphasize that, unlike the baselines, our approach avoids a major bottleneck: parameter optimization for expressions. A key advantage of our method is that it learns a posterior over both expressions and parameters, eliminating the need for computationally expensive optimization procedures.

# E ADDITIONAL RESULTS

## E.1 STROGATZ BENCHMARK

Figure 7a shows the exact symbolic recovery rate on the Strogatz benchmark (see subsection B.1). PhySO is missing from the figure since they don't report their results on that benchmark. We can see that ERRLESS achieves the highest recovery rate compared to the baselines. Notably, ERRLESS stays robust even in the noisiest setting (10%) compared to DSR, a deep reinforcement-learning based approach and BSR a bayesian method.

## E.2 BLACKBOX BENCHMARK

Figure 7b shows the mean of the median test set $R^2$ over a subset of the Blackbox benchmark (see subsection B.1. The results for competing baselines are taken from the SRBench repository [4]. Note that we run ERRLESS on the same three noise levels as the other benchmarks whereas SRBench doesn't run the baselines on any noisy versions of the benchmark. Nevertheless, our approach remains competitive and even beats closely related methods such as DSR and BSR.

---

[4] https://github.com/cavalab/srbench/blob/master/results/black-box_results.feather

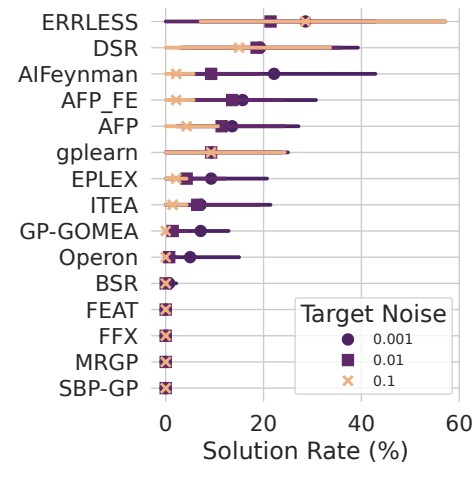
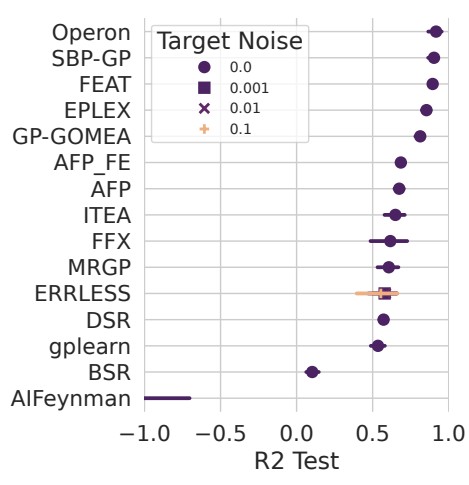

(a) Exact symbolic recovery rate for the Strogatz benchmark.

(b) Mean of the median test $R^2$ over the Blackbox benchmark.

Figure 7: **Results on the Blackbox and Strogatz benchmarks:** (a) Exact symbolic recovery rate on the Strogatz benchmark and (b) Mean of the median test $R^2$ over the Blackbox benchmark. Note that ERRLESS is benchmarked on the target noise levels $\{0.001, 0.01, 0.1\}$ whereas the baselines are only benchmarked on target noise level 0.

