# OpenReview forum: "Bayesian Symbolic Regression with Entropic Reinforcement Learning"
_ICLR.cc/2026/Conference — Submitted to ICLR 2026_

### Official Review · Reviewer_VK1w · 2025-10-29

**Soundness:** 3
**Presentation:** 2
**Contribution:** 2
**Rating:** 6
**Confidence:** 3

**Summary:**

This paper study the a Bayesian symbolic regression (SR) framework and proposes  Entropy-Regularized Reinforcement Learning for Expression Structure Sampling

ERRLESS models a posterior distribution over expressions using maximum-entropy reinforcement learning (RL). The core idea is to amortize posterior sampling of symbolic expressions using a neural policy trained via trajectory balance (a GFlowNet-style objective).

**Strengths:**

- Solid mathematical definition. The paper provides formal definitions of priors, likelihoods, unit constraints, and posterior training objectives.
- Novel idea on Translating Bayesian symbolic regression into a maximum-entropy RL setting.

**Weaknesses:**

1. **Organization and Focus of the Paper**

   * The paper’s organization is not well-balanced. A large portion of the text focuses on the details of post-order traversal and unit constraints, which could be greatly simplified or moved to the appendix.
   * Instead, the section on translating Bayesian symbolic regression into a **maximum-entropy reinforcement learning (RL)** framework should be expanded and clearly explained.
   * The motivation and benefits of using **maximum-entropy RL** are not adequately discussed — the authors should explicitly explain why this formulation is preferable compared to standard RL or other probabilistic inference methods.

2. **Comparison to Deep Symbolic Regression (DSR)**

   * The authors should more clearly differentiate their approach from **Deep Symbolic Regression**, which also employs RL and includes an entropy regularizer. It is unclear what the theoretical or practical advantage of the proposed method is relative to DSR.

3. **Clarification of Prior Work Description**

   * The statement that existing methods “have the goal of finding a single best expression” is inaccurate. Many symbolic regression methods explicitly search for a **Top-K set** or a **Pareto front** of optimal expressions under different objectives (e.g., accuracy and simplicity). The description of prior work should be corrected accordingly.

4. **Comparison with Monte Carlo Tree Search-based Methods is missing.**

5. **Improvement of Figure 1**

   * Figure 1 currently illustrates only the internal mechanism of the proposed approach. It should also highlight the **advantages or improvements** of this method compared to existing baselines — for example, how it better models uncertainty, enforces constraints more effectively, or improves sampling efficiency.

**Questions:**

- Could you clarify the conceptual and practical differences between deep reinforcement learning with entropy regularization and maximum-entropy reinforcement learning (MaxEnt RL)?
- What concrete advantages did you observe in your experiments when using MaxEnt RL compared to standard RL with entropy regularization
- What underlying factors contribute to this improvement?

---

> ### Author Response · Authors · 2025-11-20
>
> We thank the reviewer for their useful comments.
>
> For the novelty aspect and conceptual/practical differences between our approach and existing deep-reinforcement-learning-based approaches for symbolic regression, we kindly refer the reviewer to the official comment we posted.
>
> **Paper organization**: We thank the reviewer for this important feedback. We will clarify the differences in our formulation compared to the literature in the updated manuscript.
>
> **Clarification of prior work**: We will correct the prior work to accurately reflect your point.
>
> **Comparison with MCTS approaches**: The recent paper [1] uses MCTS for symbolic regression. We are currently running their algorithm on the AI Feynman dataset and will share the results by the end of the discussion period.
>
> **Improvement of Figure 1**: We thank the reviewer for their feedback, we will update the figure accordingly.
>
> [1] Huang et al. 2025. *Improving Monte Carlo Tree Search for Symbolic Regression*.

---

> ### Comment · Reviewer_VK1w · 2025-11-20
> **can you update the revised PDF during the rebuttal session?**
>
> Dear Author,
> It would be easier for me to confirm your revisions during this rebuttal period, after you have uploaded the revised PDF. Additionally, you can highlight the edited parts in a different text color.

---

> > ### Author Response · Authors · 2025-11-21
> >
> > Dear reviewer, thank you for your prompt response. We re-uploaded the manuscript by highlighting the added part in blue. We take this opportunity to emphasize that the new changes can be found in section D and E of the appendix. Changes pertaining to the figure, reorganization of the methodology and prior work are still underway and are **not currently reflected** in the updated version as we're currently working on it and will share it very soon.

---

### Official Review · Reviewer_DjoN · 2025-11-01

**Soundness:** 3
**Presentation:** 3
**Contribution:** 2
**Rating:** 2
**Confidence:** 5

**Summary:**

The paper introduces ERRLESS, a Bayesian symbolic regression framework that learns to sample both expression structures and parameters through an entropy-regularized reinforcement learning policy. Instead of finding a single best-fit formula, the method models a posterior distribution over symbolic expressions, aiming to capture uncertainty and improve robustness. It also proposes a bottom-up tree-generation process to enforce dimensional consistency of physical units. Experiments on synthetic and Feynman datasets show competitive symbolic recovery and strong predictive performance under noise.

**Strengths:**

Originality

The paper provides a coherent Bayesian framing of symbolic regression, formulating it as joint inference over expression structures and parameters and linking entropy-regularized reinforcement learning with posterior sampling.

While the use of a neural policy and RL training is standard, the integration of a Bayesian interpretation for uncertainty-aware symbolic regression offers a conceptually unified perspective that may help reframe how probabilistic ideas are applied in interpretable model discovery.

The paper also emphasizes dimensional consistency and structured priors in the generation process, aligning symbolic regression more closely with physical reasoning tasks.

Quality

The methodology is technically consistent and implemented carefully, combining entropy-regularized RL, structural constraints, and parameter sampling in a logically sound way.

The empirical section is comprehensive, evaluating ERRLESS on both synthetic and Feynman datasets with appropriate metrics (symbolic recovery, predictive accuracy, and uncertainty calibration).

The results show competitive performance and credible uncertainty estimates, demonstrating that the framework is functional and not merely theoretical.

Clarity

The paper is clearly written and well organized, with intuitive explanations, clean mathematical notation, and helpful figures.

The connection between the Bayesian posterior objective, the entropy-regularized RL formulation, and the sampling procedure is explained transparently, making the framework accessible to both symbolic regression and probabilistic ML audiences.

**Weaknesses:**

Originality and Positioning

The claimed methodological contributions are not sufficiently novel relative to existing symbolic regression frameworks. Using a neural policy trained via reinforcement learning to generate symbolic expressions has been extensively explored in prior works such as Deep Symbolic Regression (Petersen et al., 2019), the Finite Expression Method (Liang et al., 2022), PhySO (Tenachi et al., 2023), Neural-Guided Genetic Programming (Li et al., 2023). ERRLESS largely follows this paradigm, differing mainly in applying a Bayesian interpretation.

The paper could improve by clearly distinguishing what aspects of the Bayesian formulation are algorithmically new (e.g., how entropy regularization meaningfully approximates posterior inference beyond standard policy exploration) and by acknowledging that probabilistic sampling of expressions is already a common practice in symbolic regression.

The introduction and related work should also discuss Parsing the Language of Expression (Huang et al., 2025), which incorporates domain-aware symbolic priors, and other grammar-based Bayesian approaches (e.g., Probabilistic Regular Tree Priors, Grammar-Guided GP). This would provide a more accurate contextual positioning.

Bayesian Formulation and Computational Feasibility

The proposed Bayesian approach introduces significant computational overhead without evidence of efficiency or scalability benefits.

Modeling a posterior over both expression structures and parameters entails sampling from a high-dimensional, multimodal distribution, which is computationally intractable for expressions with many constants. In practice, the method collapses this to a simple Gaussian approximation, undermining the Bayesian claim.

The paper could improve by (a) including runtime and complexity analyses, (b) providing ablation studies showing how posterior sampling scales with model size, and (c) discussing approximations (e.g., variational or factorized posteriors) that might make the Bayesian framework practical.

Bottom-Up Generation Justification

The claimed advantages of bottom-up tree generation—namely producing valid intermediate expressions and easier dimensional checks—are not unique to this design.

Top-down generation frameworks can also ensure dimensional compatibility through type constraints or operator masking (as in PhySO) and can extract valid subexpressions from partial trees.

To strengthen this component, the authors should provide comparative experiments or ablation results demonstrating concrete benefits (e.g., reduced invalid expressions or faster convergence) of bottom-up versus top-down generation.

Experimental Scope and Analysis

The quantitative results show comparable or slightly worse performance than existing methods (e.g., 44% recovery on Feynman versus ~59% for PhySO and 50–60% for DSR/PySR). Yet, the discussion emphasizes conceptual rather than numerical advantages.

The paper would be stronger with statistical analyses (variance, confidence intervals) and runtime comparisons across methods, which would help clarify whether ERRLESS provides any tradeoff between accuracy, uncertainty, and efficiency.

Additionally, more real-world or scientific case studies (beyond synthetic and Feynman benchmarks) would help demonstrate the claimed benefits in uncertainty quantification or robustness.

Use of Structural and Physical Priors

The inclusion of structural priors and physical constraints is not new. Prior work such as Parsing the Language of Expression (Huang et al., 2025), SciMED (Nature 2024), ParFam (OpenReview 2025), and Grammar-Guided GP already incorporate these ideas.

The authors could strengthen their contribution by proposing a more formal or learnable prior mechanism (e.g., data-driven estimation of structural likelihoods or adaptive physical-unit embeddings) rather than manually specified constraints.

Presentation and Scope of Claims

While the writing is generally clear, several claims (e.g., “amortizes sampling of both structures and parameters efficiently” and “bottom-up generation enables better physical reasoning”) are overstated relative to the evidence.

The authors should moderate such claims or substantiate them with direct quantitative or ablation evidence. Clarifying the limitations of the Bayesian approximation would improve credibility.

**Questions:**

Clarification on Bayesian Inference Approximation

The paper frames ERRLESS as performing Bayesian inference over both structures and parameters. Could the authors clarify how the high-dimensional posterior over constants is represented and sampled in practice?

Is the policy sampling parameters from a fixed Gaussian, or is there an adaptive approximation (e.g., variational or amortized posterior)?

A more detailed description of the Bayesian approximation would help assess whether ERRLESS meaningfully captures parameter uncertainty or simply adds stochasticity to RL sampling.

Computational Complexity and Runtime Evidence

The paper claims that ERRLESS amortizes parameter fitting into policy training, implying computational advantages. Could the authors provide runtime comparisons (e.g., training time or sample efficiency) versus existing methods such as DSR, PySR, PhySO, or FEX?

How does the method scale with expression depth, number of constants, or dataset size? Concrete complexity or scaling curves would make the efficiency claim more convincing.

Bottom-Up Generation vs. Top-Down Approaches

The authors argue that bottom-up tree construction ensures valid intermediate expressions and easier enforcement of dimensional consistency.

Could they provide an empirical comparison or ablation study showing how bottom-up generation affects search efficiency, validity rates, or symbolic recovery compared to a top-down baseline?

If both strategies are possible, under what conditions is bottom-up generation more advantageous?

Quantitative Analysis of Uncertainty Quality

The paper highlights uncertainty quantification as a strength. Could the authors provide quantitative metrics for uncertainty calibration (e.g., negative log-likelihood, coverage probability) and compare them with ensemble-based or bootstrapped SR methods?

Demonstrating that the posterior uncertainty is useful (e.g., for model selection or active learning) would significantly strengthen the paper’s claims.

Evaluation Breadth and Reproducibility

The benchmarks are limited to synthetic and Feynman datasets. Are there plans to test ERRLESS on real-world scientific or engineering problems, where uncertainty estimation might be critical?

Providing open-source code and pre-trained policies would also enhance the paper’s impact and reproducibility.

Treatment of Structural and Physical Priors

The use of structural and dimensional priors resembles earlier approaches such as Parsing the Language of Expression (Huang et al., 2025), PhySO (Tenachi et al., 2023), and ParFam (2025).

Could the authors clarify what is new in their prior formulation—does it introduce learnable or data-driven components beyond manually specified grammar rules?

If possible, an ablation comparing ERRLESS with and without these priors would help quantify their contribution.

Interpretation of “Amortized Sampling”

The phrase “amortizing sampling of both expression structures and parameters” is central to the paper’s narrative. Could the authors more precisely define this term?

Is the amortization referring to shared policy parameters across different expressions (as in DSR) or to efficient reuse of parameter inference results? A clearer explanation would clarify the claimed efficiency benefits.

Positioning Relative to Prior Work

The paper would benefit from a clearer comparison to Bayesian symbolic regression literature (e.g., Probabilistic Regular Tree Priors, Grammar-Guided GP, and Neural-Guided GP).

Could the authors explain how ERRLESS differs conceptually or computationally from these methods and whether the probabilistic formulation offers tangible new capabilities?

---

> ### Author Response · Authors · 2025-11-20
>
> **Novelty**:
> While probabilistic sampling of expressions is not new, our work, to the best of our knowledge is the first to learn a policy that, at optimality, samples from a Bayesian posterior over expressions. We refer the reviewer to the comment regarding novelty for theoretical and empirical justifications.
>
> **Introduction and related works**
> We thank the reviewer for providing additional references, we will incorporate them into the manuscript.
>
> **Runtime**
> We updated the manuscript to include a runtime comparison in Figure 6 in the appendix (PhySO is missing since the paper in question does not report runtime) showing that ERRLESS is the second fastest symbolic-regression method. Note that DSR is an order of magnitude slower than our method and that's mainly due to the fact that our method doesn't need to optimize the parameters of an expression since it learns the posterior over expressions **and** parameters. The parameter optimization step is usually the most expensive part for symbolic regression methods.
>
> We would also like to highlight that ERRLESS amortizes the cost of sampling from a distribution into that of training the neural policy which at inference time would provide samples from the posterior. MCMC approaches, on the other hand, are expensive to draw samples from since they can suffer from slow mode mixing, and one would need to run MCMC chains that may take a long time to cover the target distribution every time one needs samples from the posterior.
>
> **Bottom-up generation**
> PhySO is a top-down generation method that allows for enforcing physical unit constraints; however, bottom-up generation allows a straightforward and easy way to enforce these constraints at each step, since the leaf nodes are sampled first and thus we know the units and only need to mask the operators accordingly. In top-down generation, the units of each node in an expression tree are not known until the entire subtree below is sampled. We can see from Algorithm 1 in the PhySO paper that even though they generate expressions top-down, they are forced to do a bottom-up assignment in order to accurately compute the unit constraints.
>
> Another benefit to bottom-up generation is that unlike top-down generation, one can compute rewards for intermediate states, whereas in top-down generation, an intermediate expression has "holes" and can never be evaluated unless at the end. This means that we can apply reward shaping approaches for bottom-up generation and make use of the intermediate signal to guide generation.
>
> **Use of structural and physical priors**
> We acknowledge that the use of structural and physical priors is not new, nor do we claim that as a contribution. We decided for simplicity to settle for the unigram prior, which was sufficient for achieving strong results. Exploring alternative priors would however be an interesting future direction.
>
> **Thank you for the helpful comments, and we hope that these responses and additional experiments have addressed your concerns.**

---

> > ### Comment · Reviewer_DjoN · 2025-11-28
> >
> > Partially addressed or unresolved points:
> >
> > 1. Novelty and positioning remain overstated.
> >
> > The rebuttal asserts that ERRLESS is “the first” to learn a policy that samples from a Bayesian posterior but does not provide clear algorithmic distinctions from prior probabilistic SR (e.g., PRTP, grammar-guided Bayesian SR, NGGP with stochastic sampling). The Bayesian framing still appears conceptually incremental rather than fundamentally new.
> >
> > 2. Bayesian posterior approximation remains unclear.
> >
> > The rebuttal repeats that the learned policy “samples from a Bayesian posterior,” but does not explain:
> > a. how the high-dimensional posterior over constants is actually represented,
> > b. whether the Gaussian parameter sampling is variational, amortized, or fixed, or
> > c. how approximation errors propagate.
> > This central theoretical point remains insufficiently justified.
> >
> > 3. Runtime claims need stronger evidence.
> > The runtime plot is appreciated, but:
> > a. PhySO is omitted (no runtime available),
> > b. no comparison is shown for parameter-fitting overhead in DSR/PySR under equivalent setups, and
> > c. scaling with expression depth, number of constants, or dataset size is still not analyzed.
> > The efficiency claims remain only partially substantiated.
> >
> > 4. Bottom-up vs. top-down advantages are argued but not demonstrated.
> > The rebuttal provides conceptual justifications but does not include:
> > a. an ablation comparing bottom-up vs. top-down under controlled conditions,
> > b. empirical evidence showing improved validity rates, convergence, or reward shaping effectiveness.
> > Thus, the usefulness of bottom-up generation remains qualitative.
> >
> > 5. Uncertainty quantification remains weakly evaluated.
> > The rebuttal does not address the request for uncertainty calibration metrics (coverage, NLL, calibration curves) or comparisons against ensemble/bootstrapped baselines.
> >
> > 6. Limited benchmark scope.
> > The rebuttal does not commit to evaluations beyond synthetic and Feynman datasets, limiting claims about robustness and scientific applicability.
> >
> > 7. Structural and physical priors remain simplistic.
> > Acknowledging that the priors are basic is appreciated, but the rebuttal does not clarify whether ERRLESS offers any new mechanism beyond existing grammar-based SR.
> >
> > Not addressed:
> > No ablations quantifying the effect of priors, entropy regularization, or dimensional constraints.
> > No discussion of how the Bayesian parameter approximation scales with expression size.
> > No comparison to recent Bayesian SR frameworks (e.g., PRTP, PLEx, ParFam).
> > No clarification of “amortized posterior inference” beyond descriptive wording.
> >
> > Overall assessment
> >
> > While the rebuttal adds useful runtime data and clarifies several design choices, the major issues regarding originality, the strength and validity of the Bayesian formulation, empirical calibration of uncertainty, and the comparative positioning within symbolic regression remain largely unresolved.
> >
> > My overall evaluation therefore remains unchanged.
> >
> > I keep my score at 2 (reject).

---

### Official Review · Reviewer_Hh12 · 2025-11-03

**Soundness:** 2
**Presentation:** 3
**Contribution:** 2
**Rating:** 2
**Confidence:** 5

**Summary:**

The paper proposes a Bayesian formulation for reinforcement learning of symbolic regression. The idea is to construct a joint probability distribution over the expression tree and the measurement data. The logarithm of the joint probability is used as the reward, and then an entropy regularization term is added to conduct reinforcement learning. The experiments on a small manually constructed synthetic dataset and Feynman SR database demonstrate the effectiveness of the method.

**Strengths:**

1. The writing is clear.
2. the method formulation is correct.

**Weaknesses:**

1. Novelty is limited. Entropy regularization is a common technique for RL used to encourage exploration. This has already been used in the recent most related work, DSR. It is not the unique contribution of this work.

2.  The motivation is not reasonable enough. In fact, all RL based symbolic regression is learning a probabilistic expression generator. For example DSR uses an RNN to implement Eq (3) (based on the preoder traversal of the tree). The recent CADSR [1] uses transformer to implement Eq(3) based on breadth-first search.  Via RL, all these methods are learning a posterior expression tree sampler --- given the measurement data. The proposed method mainly differs in using bottom-up order for tree generation.  It does not seem clear about the advantage.

3. The empirical results are limited. At least, the proposed method should be tested on the more comprehensive SR bench database, which includes multiple datasets (including Feynman) and many black-box problems.


[1] Bastiani Z, Kirby R M, Hochhalter J, et al. Complexity-Aware Deep Symbolic Regression with Robust Risk-Seeking Policy Gradients[J]. arXiv preprint arXiv:2406.06751, 2024.

**Questions:**

see above

---

> ### Author Response · Authors · 2025-11-20
>
> We thank the reviewer for their comments. Since the novelty aspect and further experimental results are raised by other reviewers, we put them in the official comment. Thus, we kindly refer the reviewer to our response in the official comment.
>
> Below we provide a few clarifications specific to this review.
> > In fact, all RL based symbolic regression is learning a probabilistic expression generator.
>
> We agree that existing RL-based SR methods are *stochastic* in the sense that their policies all define a distribution over expression trees. However, the probabilistic nature of ERRLESS is fundamentally different: **ERRLESS is grounded in an explicit Bayesian formulation**, where both the *prior* over expressions and their parameters and the *likelihood* of the data are modeled and combined in a principled way. As discussed in **Sec. 3.2**, this leads to a unique theoretical guarantee: **the optimal policy under our objective is proportional to the exact posterior distribution over expressions and their parameters**.
>
> In contrast, methods such as DSR, PhySO, and CADSR employ stochastic policies, but their training objectives are *not* designed to learn a generative model for expressions. Their rewards primarily target data fit (with optional complexity penalties such as BIC in CADSR), and the risk-seeking policy-gradient objective explicitly bias training toward rewarding top-performing trajectories rather than distribution matching. Thus, although all these methods generate expressions stochastically, **ERRLESS is the only one whose training objective is explicitly aligned with sampling from the Bayesian posterior**.
>
>
> > The recent CADSR [1] uses transformer to implement Eq(3) based on breadth-first search.
>
> We appreciate the reviewer pointing out the connection to CADSR. We will include CADSR in our related work section and extend our experimental comparison accordingly.
>
> CADSR introduces several valuable ideas (e.g., rank-based risk-seeking gradients, dual-index positional encodings, and BIC-based rewards). However, its optimization objective remains in the *risk-seeking policy-gradient* family initiated by DSR. In contrast, ERRLESS operates in a different regime: it leverages the **trajectory balance objective from the GFlowNets framework**. Therefore, although both approaches share surface-level similarities (e.g., transformer-based policies), the contributions and theoretical underpinnings are orthogonal.
>
> **Thank you for the helpful comments, and we hope that these responses and additional experiments have addressed your concerns.**

---

### Author Response · Authors · 2025-11-20
**Response to all reviewers**

We thank the reviewers for their efforts in reviewing our paper. We answer each review separately, but in this comment we address some recurring points that have been raised.

### Theoretical and methodological comparisons

Many reviewers asked what distinguishes our work from existing reinforcement-learning-based approaches for symbolic regression, namely DSR [1] and PhySO [2].

**Similarities:** As noted by several reviewers (and discussed in our related work section), ERRLESS, DSR, and PhySO all train a neural-network policy via a reinforcement learning objective. All three incorporate mechanisms to regularize entropy. In addition, both ERRLESS and PhySO enforce unit constraints during expression-tree generation to reduce the effective search space. Because ERRLESS employs a novel *bottom-up* generation procedure, our unit-constraint mechanism required a redesigned algorithm despite this shared high-level objective.

**Differences/Novelties:** We highlight two central novelties that distinguish ERRLESS from prior work:

1. **Explicitly encoding an expression prior in the reward.** Both DSR and PhySO use the reward $1 / (1 + \text{NRMSE})$, where for an expression tree $(T, \theta)$ with evaluation function $f_{T,\theta}$,

    $$
    \text{NRMSE}(\text{T}, \theta) = \sqrt{\frac{1}{N}\sum_{i=1}^N \frac{(y_i - f_{T,\theta}(x_i))^2}{\sigma_y^2}},
    $$

    This quantity is the square root of a negative log-likelihood under i.i.d. Gaussian noise with variance $\sigma_y^2$. Consequently, the reward in DSR and PhySO (1) **does not have a natural probabilistic interpretation as a likelihood** and (2) **reflects only data fit and does not model any prior over equations**. While both methods use “masking priors” to invalidate grammatically or dimensionally inconsistent operators, these masks enforce syntactic constraints rather than expressing a well-specified probabilistic prior over equations.

    In contrast, the ERRLESS reward is
    $$\begin{align*}
    R(T, \theta) &= \log p(T, \theta) + \log \prod_{i=1}^N \exp\left(-\frac{(y_i - f_{T,\theta}(x_i))^2}{2\sigma^2}\right)
    \\\\&=\log p(T, \theta) + \sum_{i=1}^N \left(-\frac{(y_i - f_{T,\theta}(x_i))^2}{2\sigma^2}\right),
    \end{align*}
    $$
    which **explicitly incorporates an expression prior** $p(T, \theta)$ together with a (Gaussian) likelihood, making the posterior $p(T, \theta | \mathcal D) \propto \exp R(T, \theta)$. The full Bayesian formulation adopted by ERRLESS provides additional informative signals during search and mitigates overfitting under noisy data.

2. **Training objective and its minimizer/maximizer.** Both DSR and PhySO optimize their policies using a *risk-seeking* policy-gradient objective combined with entropy regularization. Concretely, the entropy term $\lambda_{\mathcal H}\mathcal{H}(T, \theta)$ is weighted by a parameter $\lambda_\mathcal{H}$ that controls the level to which we want the policy's samples to be diverse. Because the loss is computed only on the top–$\epsilon$ fraction of high-reward trajectories (risk-seeking conditioning) and because $\lambda_\mathcal{H}$ is freely adjustable, there is **no well-defined theoretical characterization for the maximizing policy $p^*$ of their training objectives**. We also empirically verify this in the section below.

ERRLESS optimizes the trajectory balance (TB) objective [3], which is a particular case of a path consistency objective [4,5]. The TB loss is theoretically grounded in that its minimizing policy $p^*$ is guaranteed to sample any expression $(T,\theta)$ **with probability proportional to its exponentiated reward $\exp R(T,\theta)$**. In our formulation, this reward corresponds to the log-posterior, so the learned policy serves as an amortized sampler over the posterior landscape of symbolic expressions. Moreover, existing results [5] show that the TB-optimal policy $p^*$ also solves a maximum-entropy RL objective -- specifically, the one used by DSR/PhySO without risk-seeking truncation and with $\lambda_{\mathcal H}=1$. This connects ERRLESS's training dynamics to a principled RL formulation with a fully characterized optimum.


3. **Handling of continuous parameters**. To our knowedlge, ERRLESS is the first approach to learn a posterior over scalar parameters appearing in expressions as well as the expression structure. All existing deep-reinforcement-learning-based approaches find the best fit parameters for a given expression, which can incur a significant computational cost (see "Computational runtime cost" below), while MCMC methods rely on a Laplace approximation [6]. ERRLESS is able to circumvent that by learning a policy that samples the parameters.

To sum up, unlike the prior RL-based methods, ERRLESS is a principled Bayesian symbolic regression strategy, in the sense of seeking an optimal policy that samples the Bayesian posterior in a probabilistic model with a prior over expressions and i.i.d. observation noise.

(continued below)

---

> ### Author Response · Authors · 2025-11-20
> **Response to all reviewers (continued)**
>
> (continued from above)
>
> ### Empirical comparisons
>
> Moreover, we provide experimental results on the synthetic dataset we presented in the manuscript to show that DSR does not sample from the posterior distribution. We use the same setup for ERRLESS, the only difference is in the learning objective.
>
> **Posterior predictive metrics using importance sampling on the synthetic dataset.**
> We compute the coefficient of determination of the mean of the posterior predictive distribution ($R^2_{\rm PP}$) as well as the negative log-likelihood of the test data under the posterior predictive distribution (NLL). For the table below, the posterior predictive metric mean as well as the negative log-likelihood are computed using self-normalized importance sampling on 1000 expressions, using the policy as a proposal.
>
> | Algorithm ↓ / Metric → | $R^2_{\rm PP} (\gamma=0.001)$ | NLL $(\gamma=0.001)$ | $R^2_{\rm PP} (\gamma=0.01)$ | NLL $(\gamma=0.01)$ | $R^2_{\rm PP} (\gamma=0.1)$ | NLL $(\gamma=0.1)$ |
> |---|---|---|---|---|---|---|
> | **ERRLESS** | **0.99** | **-222.23** | **0.99** | **-222.10** | **0.99** | **-225.03** |
> | **DSR** | -0.48 | 2843.60 | 0.90 | -209.83 | 0.81 | 6055.76 |
> | **Gaussian Process** | -0.30 | -15.80 | -0.31 | -16.26 | -0.40 | -40.85 |
>
> **Posterior predictive metrics using Monte Carlo on the synthetic dataset.**
> Here we compute the same quantities, but taking a simple Monte Carlo average over policy samples, instead of importance sampling, which incorporates information from the unnormalized posterior.
>
> | Algorithm ↓ / Metric → | $R^2_{\rm PP} (\gamma=0.001)$ | NLL $(\gamma=0.001)$ | $R^2_{\rm PP} (\gamma=0.01)$ | NLL $(\gamma=0.01)$ | $R^2_{\rm PP} (\gamma=0.1)$ | NLL $(\gamma=0.1)$ |
> |---|---|---|---|---|---|---|
> | **ERRLESS** | **0.61** | **-221.15** | **0.98** | **-221.75** | **0.99** | **-224.78** |
> | **DSR** | -8.21 | 2843.64 | -3.23 | -209.72 | -238846.32 | 6052.21 |
> | **Gaussian Process** | -0.30 | -15.80 | -0.31 | -16.26 | -0.40 | -40.85 |
>
> The above results are consistent with the fact that ERRLESS samples from the true posterior, while DSR neither sets or achieves this goal.
>
> [1] Petersen et al. 2021. *Deep symbolic regression: Recovering mathematical expressions from data via risk-seeking policy gradients*.
>
> [2] Tenachi et al. 2023. *Deep symbolic regression for physics guided by units constraints: toward the automated discovery of physical laws*.
>
> [3] Malkin et al. 2022. *Trajectory balance: Improved credit assignment in GFlowNets*.
>
> [4] Nachum et al. 2017. *Bridging the gap between value and policy
> based reinforcement learning*.
>
> [5] Deleu et al. 2024. *Discrete probabilistic inference as control in multi-path environments*.
>
> [6] Guimerà et al. 2020. *A Bayesian machine scientist to aid in the solution of challenging scientific problems*.
>
> ### Results on the Blackbox and Strogatz benchmarks
> We provide additional results on the Strogatz benchmark which contains 14 expressions from Ordinary Differential Equations (ODE) as well as the Blackbox benchmark that contains 122 datasets.
>
> For both benchmarks, the results **do not include** PhySO since they don't report them in their paper. Moreover, we only run our method on a **subset of 66 datasets** from the Blackbox benchmark since those were the only ones to contain **continuous variables as well as continuous targets** and as such, we report results for other baselines on that subset only. Finally, note that for the Blackbox benchmark in the SRBench paper, no noise was added whereas ERRLESS was benchmarked on the same levels of noise as for Strogatz and AI Feynman benchmarks. As such, it's hard to directly compare our method and other baselines.
>
> As we can see from Figure 7a in the updated manuscript, our approach is competitive on the Strogatz dataset and beats the baselines especially on the noisiest setting (10%). On the Blackbox benchmark however (Figure 7b), we can see that our approach trails behind some Genetic-Programming-based approaches with an average $R^2$ of $0.58$ on the $1\%$ noise setting. Note that ERRLESS was benchmarked against noisy version of this benchmark whereas competing baselines (notably DSR ($R^2=0.57$) and BSR ($R^2=0.10$)) who are the closest to our approach) are run without added noise.
>
> ### Computational runtime cost
>
> Section D in the appendix provides the runtime for all methods on the blackbox dataset (see Figure 6 in the updated manuscript).
>
> We can see that ERRLESS is the second-fastest method. While we acknowledge that hardware constraints and code optimizations influence by a large margin the runtime of methods, we emphasize that the main reason why our method is very fast is that it doesn't include parameter optimization which most symbolic regression methods use. Indeed, since we learn a posterior over both expressions and parameters, our method does not suffer from the computational burden that comes in other baseline methods.

---

### Meta-Review · Area_Chair_gaRw · 2026-01-06

**Summary:**

Reviewer Hh12 and Reviewer DjoN were most critical and to a large degree pointed out an apparent lack of novelty and a also an experimental evaluation that was simply not broad enough. Although it has to pointed out that Reviewer VK1w liked the "novel idea on translating Bayesian symbolic regression into a maximum-entropy RL setting".

**Reviewer Concerns:**

Although the rebuttal addressed to a certain degree the novelty concerns. I am not convinced this was done in a thorough enough fashion. For instance, after the rebuttal Reviewer DjoN replied and kept their assessment that the novelty and positioning of the paper is still an issue. Furthermore, the experimental evaluation has not improved sufficiently in order to alleviate the concerns by the reviewers.


One thing to note about the submission, however, is that the authors suspected the review by Reviewer DjoN to be LLM generated. The authors point out that some of the references seem to be non-existent or slightly off. Also some of the quoted experimental results are not correct. On the other hand Reviewer DjoN was the only reviewer actively engaging with the rebuttal process. My opinion is that this is a valid review and should be taken into consideration.

**Reviewer Scores:**

I do not think that any of the reviewers would have changed their scores

---

### Decision · Program_Chairs · 2026-01-26

Reject